# Whole-Body 3D Pose Estimation Based on Body Mass Distribution and Center of Gravity Constraints

**DOI:** 10.3390/s25133944

**Published:** 2025-06-25

**Authors:** Fan Wei, Guanghua Xu, Qingqiang Wu, Penglin Qin, Leijun Pan, Yihua Zhao

**Affiliations:** 1School of Mechanical Engineering, Xi’an Jiaotong University, Xi’an 710049, China; wf3117370016@stu.xjtu.edu.cn (F.W.); wuqingqiang@xjtu.edu.cn (Q.W.); qplqplqpl@stu.xjtu.edu.cn (P.Q.); panleijun@stu.xjtu.edu.cn (L.P.); zyh111@stu.xjtu.edu.cn (Y.Z.); 2The State Key Laboratory for Manufacturing Systems Engineering, Xi’an Jiaotong University, Xi’an 710054, China; 3The First Affiliated Hospital of Xi’an Jiaotong University, Xi’an 710061, China; 4State Industry-Education Integration Center for Medical Innovations, Xi’an Jiaotong University, Xi’an 710049, China

**Keywords:** deep learning, 3D pose estimation, whole-body pose, Transformer, body mass distribution, center of gravity constraints

## Abstract

Estimating the 3D pose of a human body from monocular images is crucial for computer vision applications, but the technique remains challenging due to depth ambiguity and self-occlusion. Traditional methods often suffer from insufficient prior knowledge and weak constraints, resulting in inaccurate 3D keypoint estimation. In this paper, we propose a method for whole-body 3D pose estimation based on a Transformer architecture, integrating body mass distribution and center of gravity constraints. The method maps the pose to the center of gravity position using the anatomical mass ratio of the human body and computes the segment-level center of gravity using the moment synthesis method. A combined loss function is designed to enforce consistency between the predicted keypoints and the center of gravity position, as well as the invariance of limb length. Extensive experiments on the Human 3.6M WholeBody dataset demonstrate that the proposed method achieves state-of-the-art performance, with a whole-body mean joint position error (MPJPE) of 44.49 mm, which is 60.4% lower than the previous Large Simple Baseline method. Notably, it reduces the body part keypoints’ MPJPE from 112.6 to 40.41, showcasing the enhanced robustness and effectiveness to occluded scenes. This study highlights the effectiveness of integrating physical constraints into deep learning frameworks for accurate 3D pose estimation.

## 1. Introduction

### 1.1. Background and Challenges

As a hot topic in the intersection of computer vision and artificial intelligence, human 3D pose estimation has an extremely important research status and broad application prospects. As one of the challenging problems in computer vision, it has received in-depth attention from researchers for some time. As basic research work, human pose estimation has a promoting effect on many downstream studies, including human–computer interaction [1], scene modeling [2], motion analysis [3], smart medical care [4], and virtual reality [5].

However, accurately estimating the 3D pose of the human body from a 2D image is a challenging task. Depth ambiguity under a monocular view is a major obstacle [6]. In a 2D image, only the projection positions of body keypoints at a certain angle are included, and the corresponding depth information is lost. However, since the 2D pose is a projection of the 3D pose, the same 2D pose may correspond to multiple 3D poses [7], which makes the mapping from the 2D pose to the 3D pose uncertain. The self-occlusion problem also makes pose estimation difficult. When some parts of the human body are occluded by other parts, it is difficult for vision-based methods to obtain accurate information about the occluded parts [8], thus affecting the accuracy of pose estimation. Moreover, 3D pose estimation faces many problems, such as complex backgrounds [9], illumination changes [10], and scale changes [11].

The existing methods for estimating the 3D body pose on the basis of monocular color images usually adopt two-stage solutions [12,13,14,15,16,17,18]. These methods first use an off-the-shelf 2D detector [16,19,20,21,22] to estimate the 2D pose of the human body from RGB images and then use deep learning methods [23,24,25] to lift the 2D pose in order to obtain accurate 3D pose estimation results. These methods make full use of the high precision and generalization ability of current 2D pose estimation technology and provide relatively accurate basic information for subsequent 3D pose estimation. Moreover, the two-stage methods have a relatively simple network structure and high computational efficiency, which makes them easy to implement in practical applications. However, these methods are highly dependent on the accuracy of 2D pose estimation results. When the keypoint detection is inaccurate or the keypoint position cannot be identified due to occlusion, the error is eliminated in the subsequent coordinate lifting process, thereby affecting the final 3D pose estimation results.

In addition, existing methods for estimating the whole-body 3D pose of the human body generally estimate the keypoint positions directly, lacking constraints on the keypoint positions, resulting in a loss of accuracy in the keypoint position estimation. When part of the human body is occluded, the direct regression 3D coordinate methods use the model to learn complex mapping relationships directly from the images. Due to the lack of information about the occluded part, the model has difficulty capturing the complete posture features. The method of first regressing and then lifting the coordinates cannot accurately detect the occluded keypoints, which seriously affects the final 3D posture estimation results. In addition, traditional methods do not effectively constrain the estimated positions of keypoints and make insufficient use of prior knowledge.

### 1.2. Related Work

#### 1.2.1. Transformer Structure

In the task of 3D human pose estimation, traditional methods can be roughly divided into two categories according to the network model used: convolutional neural network-based methods [2,7,26,27,28,29,30,31] and Transformer-based methods [25,32,33,34,35,36,37,38,39,40]. Convolutional neural networks can efficiently capture underlying visual features such as edge textures in images, while the weights and local connection characteristics of convolution operations make their complexity and number of parameters much lower than the Transformer structure. Moreover, convolutional neural networks expand the receptive field by stacking multiple layers of convolution, which makes it difficult to directly model the dependencies of distant joints, resulting in weak global dependency modeling capabilities. The self-attention mechanism of the Transformer structure can directly calculate the dependencies between any two keypoints and has strong global correlation modeling capabilities. Zheng et al. [37] proposed a pure Transformer framework that uses the 2D joint coordinates of multiple consecutive frames as inputs, uses a network structure to extract cross-frame temporal features and single-frame spatial features, and encodes the keypoint positions in time and space, thereby solving the depth blur and occlusion problems in video sequences. Most of the subsequent Transformer-based methods have improved upon this method. Li et al. [32] proposed a multi-hypothesis Transformer structure, which models multiple hypotheses for the same pose and selects the one with the highest credibility as the final result. Zhang et al. [34] proposed a mixed spatiotemporal Transformer model that uses the Transformer structure to alternately extract the temporal and spatial features of keypoints.

In this work, we propose using the Transformer structure to model the position encoding of keypoints, use the change in the center of gravity position and posture to establish a mapping relationship, and use the mass distribution of body parts to constrain the estimated position of each keypoint, so that the model can better understand the structural limitations of the human pose, thereby improving the authenticity and effectiveness of the pose estimation results.

#### 1.2.2. Whole-Body 3D Pose Estimation

Existing whole-body 3D human pose estimation methods can be divided into end-to-end methods [41,42,43,44,45,46] and multistage methods. End-to-end methods directly regress 3D coordinates from 2D joint points and build a general whole-body pose estimation model on the basis of data-driven explicit priors. Wen et al. [41] used a semantic graph attention network to model the semantic relationships between joints and used the distance information between the joints as auxiliary supervision to enhance the rationality of the pose. Zhu et al. [42] proposed the fine annotation of the keypoints of the whole body and combined the Transformer structure to model video sequence dependencies and geometric constraints to improve the consistency of body and hand movements. Semet et al. [45] divided the human body into multiple parts, modeled different parts independently, and introduced a temporal attention mechanism to extract the motion patterns of adjacent frames in the feature extraction process of spatiotemporal sequences, improving the adaptability of the model to complex scenes.

The multistage method [47,48,49,50,51,52,53] divides the keypoints of the whole body into body parts, faces, and hands, estimates them separately, and gradually refines them through a cascade network. Rong et al. [47] proposed dividing the human body into three independent modules and performed posture regression separately. Moreover, the output of each module is integrated through coordinate transformation and optimization fitting to ensure the rationality of the movement. Weinzaepfrl et al. [49] proposed an integrated framework based on knowledge distillation, which divides keypoints into body, hand, and face, builds expert models for each keypoint for training, and unifies them into the global model. This method is robust under complex backgrounds, occlusions, and lighting changes, but the model has a large number of parameters, and the quality of the local expert model directly affects the performance of the global model.

In this work, we propose a whole-body 3D pose estimation method based on weight distribution and center of gravity constraints. Based on the human body structure, we use the network to establish the mapping relationship between different postures and center of gravity positions and use the body mass distribution relationship to constrain the estimated positions of keypoints. Considering the ambiguity of depth information, multiple encoding methods are proposed to encode the keypoint positions, and the most similar result is selected from multiple estimation results as the final prediction result to improve the estimation accuracy of the whole-body 3D pose.

### 1.3. Research Gap and Our Contributions

Existing methods directly regress joint positions without enforcing anatomical plausibility, resulting in physically unreasonable poses, such as limb penetration or body proportion distortion under occlusion or depth ambiguity. At the same time, existing methods make insufficient use of prior information and do not fully consider the biomechanical properties of human kinematics when estimating poses, resulting in occlusion sensitivity.

To bridge these gaps, we propose a method for estimating the whole-body 3D pose on the basis of the body mass distribution and center of gravity constraints. Based on the limb connection relationship and the Transformer structure, the positions of the keypoints are encoded, and a mapping relationship between the different postures and center of gravity positions is established. The estimated positions of the keypoints are constrained by the mass relationship of each body segment so that the model can better understand the physical limitations of human poses, thereby improving the authenticity and effectiveness of the pose estimation results.

Our contributions to the whole-body 3D pose estimation include the following aspects:We propose a new method for whole-body 3D pose estimation on the basis of the body mass distribution and center of gravity position constraints, and our method achieves the SOTAs on a widely used dataset.Based on the observation of different postures and weight changes in the human body and the research of existing methods, the changing rules of the center of gravity of the human body under different postures are fully considered, and the prior information in the process of estimating the human body posture is increased in the network, which effectively solves the problem of the inaccurate estimation of keypoint positions due to occlusion in the posture estimation process.On the basis of the mass distribution relationship of human body segments, we add body mass distribution constraints when the network is used to estimate the whole-body poses, establish a mapping relationship between the center of gravity position and the human body pose, and use two different methods to encode the keypoint position information, thereby improving the accuracy of pose estimation.

## 2. Methodology

Inspired by the above methods, we propose a method for estimating the whole-body 3D pose on the basis of the body segment mass distribution and center of gravity constraints. We use the Transformer structure to model the relationships between human keypoints and encode the keypoint positions in two different ways. We use the body segment mass distribution relationship to estimate the center of gravity position and use the network to model the mapping relationship between the center of gravity position and the pose. Figure 1 shows the overall framework of our proposed method. The input is 2D keypoints from the detector, bone vector calculation and bone position calculation are used as constraints, the Transformer with the cross-attention mechanism is the main network framework, the weight is estimated through mass distribution, and a loss function based on keypoint distribution and the center of gravity position is designed.

### 2.1. Keypoint Position Encoding Based on the Human Body Structure

Under the camera imaging principle of the pinhole imaging model [54], the three-dimensional coordinates P(x,y,z) of a point in the scene are related to the projection of its pixel coordinate point J(u,v) in the image, as expressed in Equation (1):(1)$$\left\{\begin{array}{c} \frac{x}{u}=\frac{z}{f_{x}}\\ \frac{y}{v}=\frac{z}{f_{y}} \end{array}\right.$$

In (1), $f_{x}$ and $f_{y}$ represent the focal length of the camera lens. The camera imaging principle based on the pinhole imaging model is shown in Figure 2.

According to the principle of constant limb length, the two joint points have the following relationship, as expressed in Equation (2):(2)$$\left\{\begin{array}{c} \sqrt{{\left(\Delta_{x}\right)}^{2}+{\left(\Delta_{y}\right)}^{2}+{\left(\Delta_{z}\right)}^{2}}=L\\ \Delta_{x}=u_{1}\cdot \frac{z_{1}}{f_{x}}-u_{2}\cdot \frac{z_{1}+\Delta_{z}}{f_{x}}\\ \Delta_{y}=v_{1}\cdot \frac{z_{1}}{f_{y}}-v_{2}\cdot \frac{z_{1}+\Delta_{z}}{f_{y}} \end{array}\right.$$

In (2), $\Delta_{z}$ represents the vertical coordinate position error of keypoints $J_{1}$ and $J_{2}$. According to the above equation, since the length of the human limbs remains unchanged, the positions of the keypoints in different coordinate systems can be converted to each other via the above equation.

When a coordinate system is used to represent the position of keypoints, due to the limitations of image pixels and resolution, precision loss occurs when the keypoint positions in the real world are mapped to the image pixel coordinate system, resulting in deviations in the real keypoint positions. To reduce this deviation, this method uses a bone vector method to represent the keypoint positions. First, the keypoints of the human body are modeled according to the human body structure and simplified into a tree structure, with the keypoint at the human pelvis as the root node and other keypoints expanding outward from the root node [55]. The simplified human body model is shown in Figure 3.

As shown in Figure 3, the human body is connected by multiple joints and bones, which can be regarded as a tree structure with the center of the pelvis as the starting point. On the basis of the above connection relationship, this method uses the bone vector representation to encode the position of each keypoint. Through the bone vector representation, each keypoint can be regarded as the endpoint of a vector, and the bones connecting adjacent keypoints can be represented by a vector. The vector not only defines the length of the bone but also determines the direction and position of the bone in three-dimensional space, thereby fully describing the posture of the human body.

The position of the elbow joint is taken as an example. In coordinate notation, its coordinates are expressed as $E(x_{e},y_{e},z_{e})$, and the coordinates of the adjacent shoulder joint and wrist joint are expressed as $S(x_{s},y_{s},z_{s})$ and $W(x_{w},y_{w},z_{w})$, respectively. In the bone vector representation, the bone vector from the elbow to the shoulder is represented as $\vec{ES}=(x_{s}-x_{e},y_{s}-y_{e},z_{s}-z_{e})$, and the bone vector from the elbow to the wrist is represented as $\vec{EW}=(x_{w}-x_{e},y_{w}-y_{e},z_{w}-z_{e})$. Moreover, the length of the upper arm between the shoulder joint and the elbow joint can be expressed in Equation (3):(3)$$L_{ua}=\left|\vec{ES}\right|$$

In (3), $\vec{SE}$ is the bone vector from the elbow to the shoulder. And the length of the forearm between the elbow joint and the wrist joint can be expressed in Equation (4):(4)$$L_{fa}=\left|\vec{EW}\right|$$

In (4), $\vec{EW}$ is the bone vector from the elbow to the wrist. So the angle of the elbow joint can be expressed in Equation (5):(5)$$\theta_{e}=\arccos\frac{\vec{ES}\cdot \vec{EW}}{L_{ua}\cdot L_{fa}}$$

In (5), $\vec{ES}$ and $\vec{EW}$ are the bones vectors from the elbow to the shoulder and wrist, respectively. And the $L_{ua}$ and $L_{fa}$ are the lengths of the corresponding bone vector.

We use the Transformer structure to encode the two representation methods into feature vectors. For the encoding of the coordinate point representation method, each coordinate point $p_{i}$ is linearly embedded as expressed in Equation (6):(6)$$e_{i}^{p}=W_{p}p_{i}+b_{p}$$

In (6), by adding a learnable positional encoding, its topological order can be preserved. For the encoding of the bone vector representation method, the length and direction of each bone vector encoder are as expressed in Equation (7):(7)$$e_{i}^{b}=W_{b}(\frac{b_{i}}{{\left\|b_{i}\right\|}_{2}}⊕\log({\left\|b_{i}\right\|}_{2}+\varepsilon ))+b_{b}$$

Moreover, a dual-branch cross-attention structure is used to model the coordinate point modality and the bone vector modality to establish the global spatial dependency between joints and the local motion constraints of the bone chain to reduce posture ambiguity.

### 2.2. Body Mass Distribution and Center of Gravity Constraints

To make the keypoint estimation results more consistent with human kinematics, this method divides the human body into reasonable segments and accurately calculates the mass distribution of each segment. The body segments are divided according to the body parts as shown in Figure 4.

According to the structural characteristics of human anatomy, the human body is divided into multiple main segments, such as the head, trunk, upper limbs, and lower limbs. According to the “Inertial Parameters of Adult Human Body [56]”, we establish the connection between the keypoint position and the mass of the body segment and assign relative mass coefficients to each body segment to regard the human body model as a real physical model. The relative mass distribution of the “Inertial Parameters of the Adult Human Body” is shown in Table 1.

The center of gravity is the balance point of the body’s mass distribution. Corresponding to the structure of the human body, the position of the center of gravity also changes at any time. Since each part of the human body can be regarded as a rigid body, the position of the center of gravity can be obtained by the algebraic sum of the moments of each part. Therefore, we use the moment synthesis method to determine the center of gravity position and establish a mapping relationship between the different postures of the human body and the center of gravity position.

In the human anatomy model, the human body is divided into different parts. The part close to the heart is called the proximal segment, and the part far from the heart is called the distal segment. Each part is regarded as an independent rigid body model. Taking the right upper arm as an example, the proximal end of the segment is the right shoulder keypoint, and the distal end of the segment is the right elbow keypoint. The center of gravity position is shown in Figure 5.

Multiply the right shoulder joint coordinates by the corresponding proximal coefficient and the right elbow joint coordinates by the corresponding distal coefficient to obtain the three-dimensional center of gravity coordinates of the right upper arm as expressed in Equation (8):(8)$$\left\{\begin{array}{c} x_{rua}=x_{rs}\cdot l_{p}+x_{re}\cdot l_{d}\\ y_{rua}=y_{rs}\cdot l_{p}+y_{re}\cdot l_{d}\\ z_{rua}=z_{rs}\cdot l_{p}+z_{re}\cdot l_{d} \end{array}\right.$$

In (8), $\left(x_{rua},y_{rua},z_{rua}\right)$ represents the three-dimensional center of gravity coordinates of the right upper arm; $\left(x_{rs},y_{rs},z_{rs}\right)$ and $\left(x_{re},y_{re},z_{re}\right)$ represent the coordinate positions of the right shoulder joint and the right elbow joint, respectively. $l_{p}$ and $l_{d}$ represent the proximal coefficient and distal coefficient of the right upper arm segment, respectively. For the human body, the center of gravity position is the algebraic sum of the center of gravity positions of each body segment. That is, the three-dimensional center of gravity coordinate position of the human body as a whole is expressed in Equation (9):(9)$$\left\{\begin{array}{c} X_{CoG}=\sum_{i}k_{i}x_{i}\\ Y_{CoG}=\sum_{i}k_{i}y_{i}\\ Z_{CoG}=\sum_{i}k_{i}z_{i} \end{array}\right.$$

In (9), $\left(X_{CoG},Y_{CoG},Z_{CoG}\right)$ represents the three-dimensional center of gravity coordinate position of the human body as a whole, $\left(x_{i},y_{i},z_{i}\right)$ represents the three-dimensional center of gravity coordinate position of each body segment, and k represents the mass coefficient of the body segment. To obtain the accurate center of gravity position, this paper obtains the corresponding body segment mass coefficient on the basis of the relative mass distribution of each body segment in the “Inertial Parameters of Adult Human Body”, as shown in Table 2.

### 2.3. Loss Function

To train and optimize the model, it is necessary to design the corresponding loss function. In the traditional method of human 3D pose estimation, the commonly used loss function is the average joint position error loss function, and its functional form is expressed in Equation (10):(10)$$L=\frac{1}{N}\sum_{i=1}^{N}{\left\|p_{i}-{\hat{p}}_{i}\right\|}_{2}$$

In Equation (10), $p_{i}$ and ${\hat{p}}_{i}$ represent the predicted position and ground truth position of the keypoints, respectively. N represents the total number of keypoints.

This method uses the weight distribution relationship and the center of gravity position to constrain the position of the keypoint estimation. Because of the mapping relationship between different postures and the center of gravity position, we also designed the corresponding loss function as expressed in Equation Equation (11):(11)$$L_{gd}=\frac{1}{N}\sum_{i=1}^{N}{\left\|g_{i}-{\hat{g}}_{i}\right\|}_{2}$$

In Equation (11), N represents the number of all keypoints, $g_{i}$ represents the distance from the predicted keypoint to the predicted center of gravity, and ${\hat{g}}_{i}$ represents the distance from the ground truth keypoint to the ground truth center of gravity.

Moreover, the invariance of the limb length is used as a constraint to design the loss function. In the bone vector representation method, for a certain keypoint, two adjacent keypoints and their angles are used to represent the current keypoint, so the consistency of the limb length and the cosine similarity of the angle are used as the measurement criteria, as expressed in Equation Equation (12):(12)$$L_{bone}=\frac{1}{N-1}\sum_{i=1}^{N-1}\left(1-\frac{b_{i}\cdot {\hat{b}}_{i}}{\left\|b_{i}\right\|\left\|{\hat{b}}_{i}\right\|}\right)+\lambda {\left\|\left|b_{i}\right|-\left|{\hat{b}}_{i}\right|\right\|}_{2}$$

In (12), N represents the total number of keypoints, $b_{i}$ represents the predicted limb vector, and ${\hat{b}}_{i}$ represents the ground truth limb vector. Therefore, the loss function used in this paper is expressed in Equation (13):(13)$$Loss_{total}=L+L_{gd}+L_{bone}$$

In this work, the weighted sum of multiple loss functions is used as the total loss function to participate in network training, and different weight parameters are used during the training process.

## 3. Experiments

### 3.1. Datasets and Evaluation Metrics

In this work, we use the Human 3.6M WholeBody [42] public dataset for network training and verification. The Human 3.6M WholeBody dataset is an annotation extension of the Human 3.6M [57] dataset. It follows the keypoint layout used in the COCO-WholeBody [58] dataset and focuses on the 3D body pose annotation and estimation tasks. On the basis of the original annotation, the detailed keypoint annotations of the face, hands, and feet are added. The Human 3.6M WholeBody dataset consists of 133 pairs of 2D and 3D whole-body keypoint annotations from a set of 100,000 images in the Human3.6 M dataset. The training set used in this paper contains 80,000 images and 2D and 3D pose triplets; the test set contains 20,000 images and 2D and 3D pose triplets.

To verify the accuracy of the pose estimation results, we use the commonly used evaluation indicator, the mean joint position error (MPJPE). This indicator refers to the average Euclidean distance error between the predicted joint points and the actual joint points, and its formula is expressed in Equation (14):(14)$$MPJPE=\frac{1}{N}\sum_{i=1}^{N}\left\|p_{i}-{\hat{p}}_{i}\right\|$$

In (14), N is the number of keypoints, and $p_{i}$ and ${\hat{p}}_{i}$ are the predicted coordinates and ground truth coordinates of the i-th joint point, respectively.

### 3.2. Implementation Details

We used NVIDIA RTX2080Ti GPUs to train and validate the network on PyTorch 1.13.1. We used an off-the-shelf 2D keypoint detector [44] and normalized the coordinates centered on the pelvis. We performed data augmentation on the input data, including random rotation (±15°), scaling (0.8–1.2), and flipping. These operations preserve anatomical constraints (e.g., limb length) while expanding the training distribution. The deep learning model configuration includes the Transformer architecture and dual-branch cross-attention. The Transformer structure has 6 layers, 8 attention heads, and a hidden layer dimension of 256. The dual-branch cross-attention has four layers for feature fusion. We used the AdamW optimizer to train the network for 210 epochs with a batch size of 64 and an initial learning rate of 0.001. We reduced the learning rate at the 170th and 200th epochs, respectively, and the learning rate decay strategy was cosine annealing.

### 3.3. Performance Comparison

Results on the Human 3.6M WholeBody dataset. Table 3 shows the prediction accuracy of different body part keypoints in the test set. Following the 2D-to-3D lifting method, we used the existing 2D whole-body keypoint pose detector and the detected 2D pose results as input to predict the 3D pose of the whole-body keypoints. As shown in Table 3, our proposed method achieves the best results in 3D pose estimation for all parts’ keypoints. Among them, the MPJPE index for the whole-body keypoints reached 22.07, which is 20.5% higher than that of the current best method. Moreover, the MPJPE index of our proposed method for the estimation of facial keypoints reached 6.09, which is 41.67% higher than that of the current best method. This is because when we perform pose estimation, we establish a mapping between different poses and center of gravity positions and use the body mass distribution to constrain the estimated position of the keypoints, thereby reducing the error between the estimated keypoints and the actual keypoint positions. Moreover, the model achieves a computational speed of 28 FPS on an NVIDIA RTX 2080Ti, with a computational cost of 12.5 GFLOPs and 38M parameters, meeting the near-real-time rendering requirement (30 FPS). This is faster than JointFormer [40] (22 FPS) and SemGAN [41] (19 FPS). This shows the real-time performance of the proposed model, providing a basis for further online applications.

### 3.4. Ablation Study

To validate the contributions of the components of our proposed method and the corresponding loss functions, we designed ablation experiments on the Human 3.6M WholeBody dataset.

#### 3.4.1. Effects of the Body Mass Distribution and Center of Gravity Position Constraints

To verify the influence of the proposed method on the whole-body 3D pose estimation on the basis of the body mass distribution and the center of gravity position constraints, we selected the methods of adding only the body mass distribution, only the center of gravity position constraints, and both for the keypoints and designed an ablation experiment on the Human 3.6M WholeBody dataset. For the sake of fairness, the other network parameters were set exactly the same, and the ablation experiment results are shown in Table 4.

Table 4 shows that for the estimation of whole-body keypoints, the MPJPE result of adding only the body mass distribution constraints is 67.5, whereas the MPJPE result of adding the body mass distribution and the center of gravity position is 44.49, which is an improvement of 34.08%. This further shows that by encoding the keypoint information via the physical structure of the human body and the center of gravity position, and using the body mass distribution relationship and the center of gravity position to estimate the keypoint position, the accuracy and robustness of the keypoint position estimation can be improved. At the same time, it can be seen from the table that the interaction between the weight and center of mass constraints is crucial: weight distribution can enhance the consistency of limb lengths, while center of mass constraints can ensure the rationality of global posture. As shown in Table 4, combining the two can reduce the MPJPE by 34.08% compared with the constraints based on weight alone, which shows that the center of mass provides a global physical prior that can complement the local constraints based on weight.

#### 3.4.2. Impact of the Loss Function

Table 5 compares the results under different loss functions. We compared the effects of different combinations of loss functions on the final pose estimation results. As shown in the table, when the center of gravity position loss function is added, the MPJPE result for whole-body keypoints is 51.75, which is 13.3% greater than the result when only the L loss function is used. When the body mass distribution loss function is added, the MPJPE result is 17.45% greater than the result when only the L loss function is used. The results once again verify that adding the body mass distribution and center of gravity position constraints contributes greatly to the 3D whole-body pose estimation.

### 3.5. Discussion

This paper proposes a new method for 3D whole-body pose estimation. The network is used to establish the mapping relationship between the center of gravity position of the human body and different poses, and different body segment masses are added as constraints to improve the estimation accuracy of the whole-body keypoints. In addition, the proposed method achieved encouraging results.

Table 1 reports the comparison results of our method with other SOTAs. We use the results of an existing 2D whole-body keypoint pose detector as input. Compared with the existing SOTAs, our proposed method has better estimation results for keypoints in different parts of the body and obtains the smallest average error. This is because we use the network to establish the mapping relationship between the center of gravity position and different poses and add the body mass distribution to constrain the estimated position of the keypoints. Compared with the SemGAN [41] method, our method uses limited human structure relationships to constrain the estimated position of the keypoints when performing pose estimation, so the prior information is not fully utilized. Our method exploits the human body structure as prior information, encodes the keypoint locations in two ways, and constrains the estimated locations of keypoints using body segment mass distribution and the center of gravity. A comparison of the results of the ablation experiment also illustrates this point. The Human 3.6M WholeBody dataset follows the official split: 80,000 training images and 20,000 test images, which aligns with prior works [42,57]. Cross-validation is not performed due to the dataset’s standardized protocol, but the model’s generalizability is validated by outperforming SOTAs on the test set. Future work will explore cross-dataset validation on the MPII human pose. Figure 6 shows some visual comparison results. It can be seen from the figure that, in addition to the small gap between the simple pose estimation result and the true value, the proposed method can still obtain a relatively reasonable 3D pose estimation result in the case of partial occlusion. This shows the effectiveness of the proposed method and also indicate the direction for subsequent methods.

In order to quantitatively analyze the improvement of the method in 3D pose estimation, this paper compares the estimation results of different poses with the existing methods on the Human 3.6M dataset, and the results are shown in Table 6. As can be seen from the table, compared with the existing methods, the method proposed in this paper has a significant improvement in the estimation of simple actions. For example, compared with the JointFormer method, the estimation accuracy of the method in walking action is improved by 29.2%, and the estimation accuracy of the method in eating action is improved by 27.9%. This is due to the addition of center of gravity position constraints and body segment mass constraints in the process of pose estimation. In some complex actions, especially in the presence of occlusion, this method can still obtain relatively reliable results. As shown in Figure 6, in the visual comparison results of this method and the true value, the red line representing the predicted result still remains close to the blue line representing the true result in the occlusion scene.

One limitation of this method is that it relies on the Human 3.6M dataset, and its generalization ability to natural scene datasets needs further verification. In wild scenes, pose estimation performance may degrade due to extreme occlusions, unseen camera angles, and low-resolution inputs. Future work will explore domain adaptation techniques to enhance its robustness in different environments. Currently, our method can only estimate the keypoints defined in the dataset in a supervised manner. Moreover, 3D pose estimation of the whole-body keypoints of a single frame cannot meet the requirements of online use. In future work, we plan to achieve the real-time estimation of full-body keypoints and further improve the accuracy of full-body keypoint pose estimation. At the same time, we will develop unsupervised/semi-supervised learning for real-time deployment and integrate motion blur enhancement techniques to enhance the robustness of the model in dynamic scenes. We will also consider exploring multimodal fusion (e.g., RGB-D) to achieve cross-domain generalization and combine it with instance segmentation for multi-person pose estimation in extended crowded scenes.

## 4. Conclusions

In this paper, we propose a new method to estimate the whole-body 3D pose on the basis of the mass distribution of body segments and the center of gravity position constraints. This method uses a network model to encode the position information of the whole-body keypoints of the human body, establishes a mapping relationship between different postures of the human body and the center of gravity position, and constrains the estimated position of the keypoints with the mass distribution of each body segment of the human body to obtain a reliable position estimate of the whole-body keypoints of the human body. We conducted experiments on the widely used Human 3.6M WholeBody dataset. The experimental results show that this method reached a new SOTA level, and the estimation error of the keypoints of different parts improved.

## Figures and Tables

**Figure 1 sensors-25-03944-f001:**
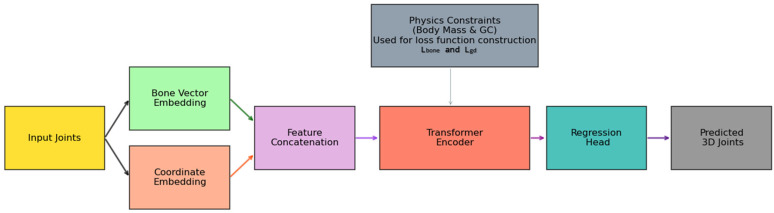
Overview framework of the proposed method. Input joints: 2D keypoints from detector → 3.1. datasets and evaluation metrics; bone vector embedding → 2.1 keypoint position encoding based on the human body structure; coordinate embedding → 2.2 body mass distribution and center of gravity constraints; Transformer with regression head → 3.2. implementation details; loss functions with physics constraints (L, L_gd_, L_bone_) → 2.3 loss function.

**Figure 2 sensors-25-03944-f002:**
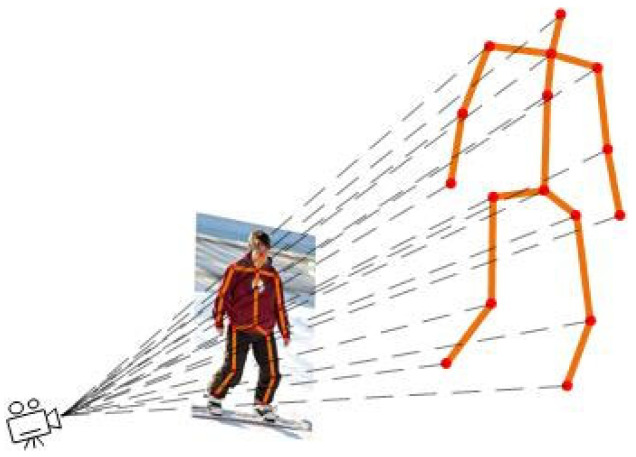
Schematic diagram of imaging principle based on the pinhole model.

**Figure 3 sensors-25-03944-f003:**
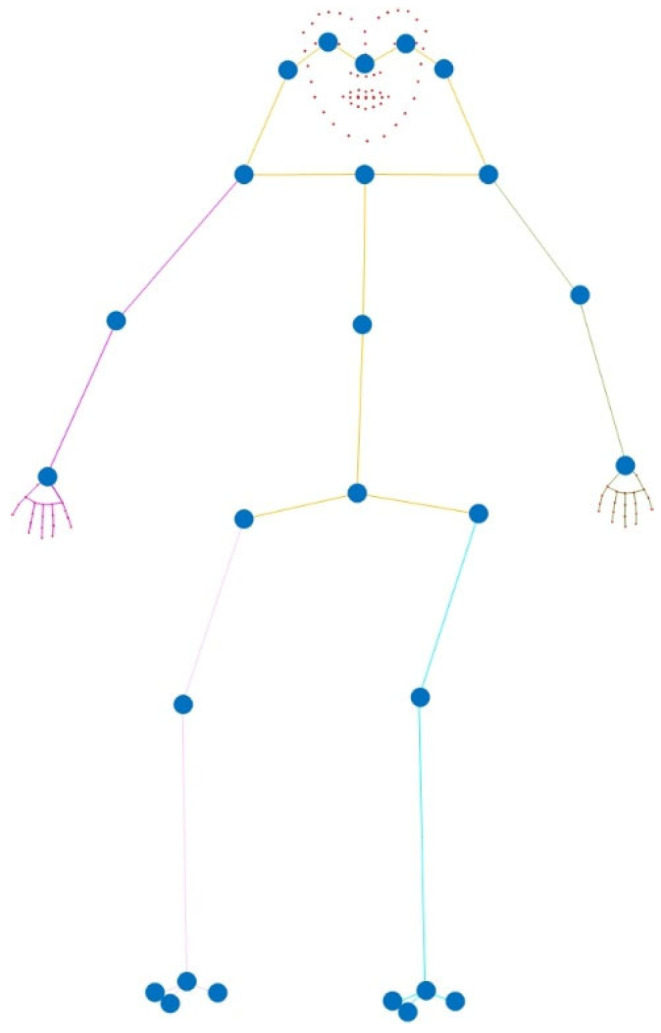
Simplified tree diagram model of whole-body keypoints.

**Figure 4 sensors-25-03944-f004:**
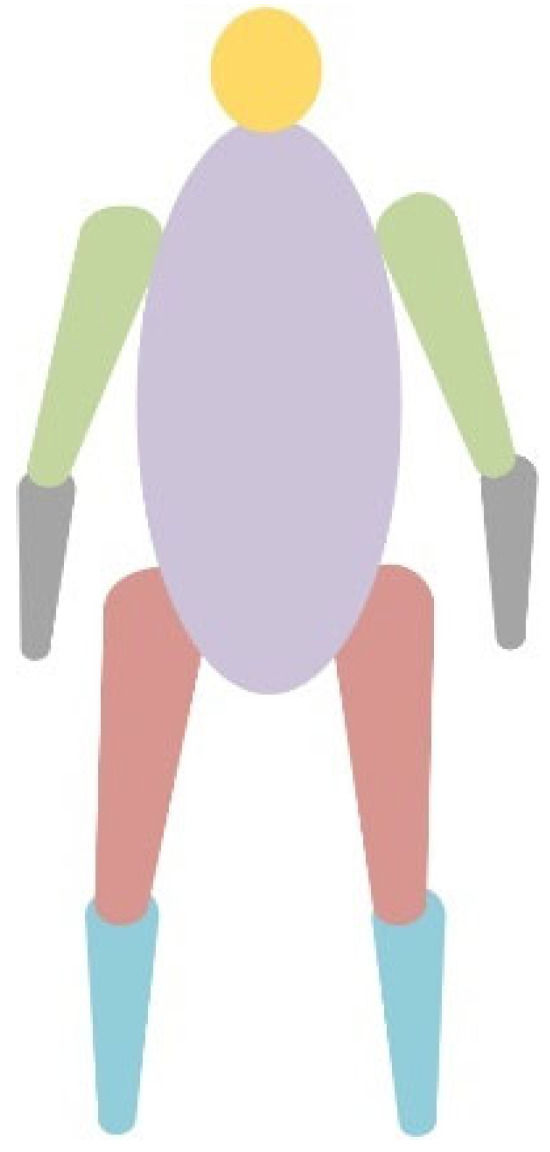
Schematic diagram of body part segmentation based on segment mass relationships. Segment division aligns with mass coefficients in Table 1, following [56].

**Figure 5 sensors-25-03944-f005:**
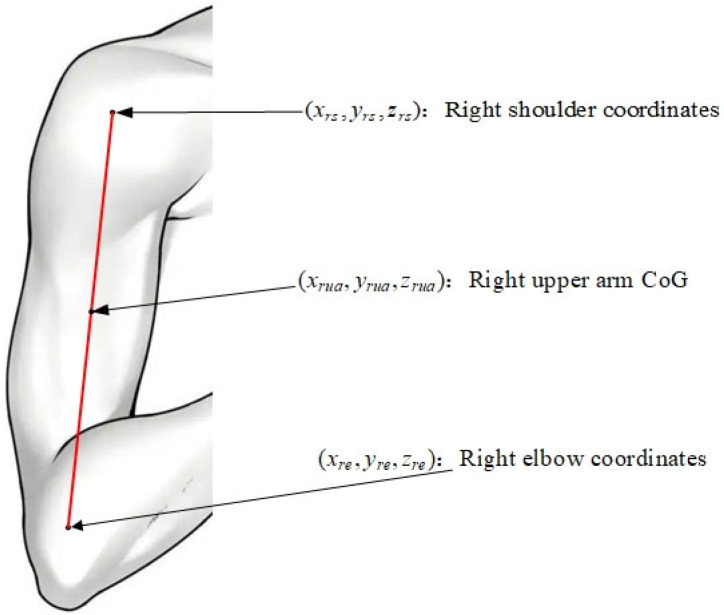
The center of gravity position of the right upper arm based on the moment synthesis method.

**Figure 6 sensors-25-03944-f006:**
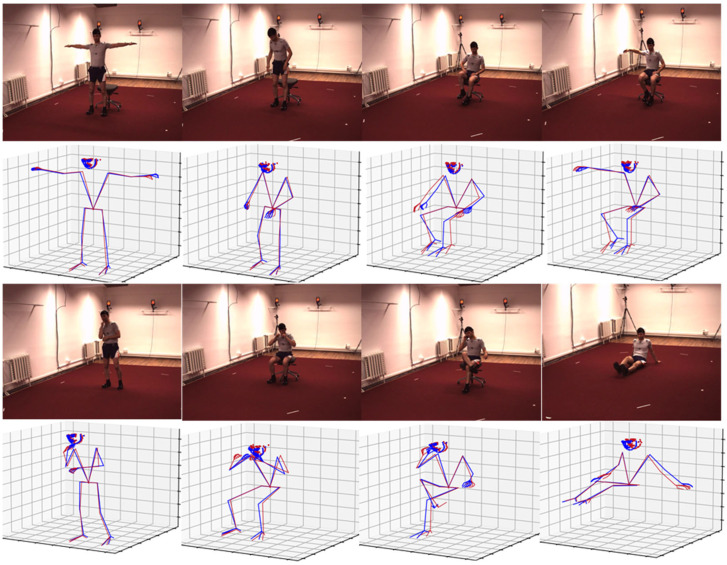
Visualization results of whole-body 3D pose estimation. The blue line represents the ground truth, and the red line represents the estimated result of our method.

**Table 1 sensors-25-03944-t001:** Relative Mass Distribution of each body segment.

Body Segment	Gender	Relative Quality/%	Body Segment	Gender	Relative Quality/%
Neck	M	8.62	Upper Arm	M	2.43
	F	8.20		F	2.66
Upper Torso	M	16.82	Forearm	M	1.25
	F	16.35		F	1.14
Lower Torso	M	27.23	Hand	M	0.64
	F	27.48		F	0.42
Thigh	M	14.19	Foot	M	1.48
	F	14.10		F	1.24
Calf	M	3.67			
	F	4.43			

**Table 2 sensors-25-03944-t002:** Relative positions of the center of mass of each body segment.

Body Segment	Gender	$L_{es}$	$L_{ex}$	Body Segment	Gender	$L_{es}$	$L_{ex}$
Neck	M	46.9	53.1	Upper Arm	M	47.8	52.2
	F	47.3	52.7		F	46.7	53.3
Upper Torso	M	53.6	46.4	Forearm	M	42.4	57.6
	F	49.3	50.7		F	45.3	54.7
Lower Torso	M	40.3	59.7	Hand	M	36.6	63.4
	F	44.6	55.4		F	34.9	65.1
Thigh	M	45.3	54.7	Foot	M	48.6	51.4
	F	44.2	55.8		F	45.1	54.9
Calf	M	39.3	60.7	Overall Centroid	M	43.8	56.2
	F	42.5	57.5		F	44.5	55.5

$L_{es}$ and $L_{ex}$ represent the percentage of the upper and lower dimensions of the centroid of each body segment to the total length of the body segment, respectively.

**Table 3 sensors-25-03944-t003:** Results for whole-body 3D pose estimation on the H3WB dataset.

Method	WholeBody	Body	Face	Hands
Simple Baseline [42]	125.4	125.7	24.6	42.5
Large Simple Baseline [21]	112.3	112.6	14.6	31.7
JointFormer [40]	88.3	84.9	17.8	43.7
3D-LFM [59]	64.13	60.83	10.44	28.22
SemGAN [41]	47.87	45.39	15.95	27.77
Ours	44.49	40.41	6.09	22.07

**Table 4 sensors-25-03944-t004:** Effects of adding body mass distribution and center of gravity position constraints.

Constraints	WholeBody	Body	Face	Hands
body mass distribution constraints	67.50	59.24	10.10	29.41
center of gravity position constraints	58.34	50.76	8.32	25.73
body mass distribution and center of gravity constraints	44.49	40.41	6.09	22.07

**Table 5 sensors-25-03944-t005:** Ablation study results for loss function.

Loss Function	WholeBody	Body	Face	Hands
L	59.71	47.69	15.35	31.93
$L_{gd}$	51.75	44.21	13.29	25.61
$L_{bone}$	49.29	45.57	9.90	26.61
$Loss_{total}$	44.49	40.41	6.09	22.07

**Table 6 sensors-25-03944-t006:** Quantitative comparison on the human 3.6M whole-body dataset.

Methods	Dir.	Disc.	Eat.	Greet	Phone	Photo	Pose	Purch.	Sit	SitD.	Smoke	Wait	WalkD.	Walk	WalkT.
PoseFormer [37]	41.5	44.8	39.8	42.5	46.5	51.6	42.1	42.0	53.3	60.7	45.5	43.3	46.1	31.8	32.2
JointFormer [40]	45.0	48.8	46.6	49.4	53.2	60.1	47.0	46.7	59.6	67.1	51.2	47.1	53.8	39.4	42.4
Ours	41.4	38.1	33.6	41.7	42.6	50.2	42.3	40.9	57.0	61.2	43.5	46.6	43.4	27.9	30.6

## Data Availability

Data are contained within the article.

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
