# Peer review of "Whole-Body 3D Pose Estimation Based on Body Mass Distribution and Center of Gravity Constraints"

_sensors, 2025, doi:10.3390/s25133944_

Round 1
Reviewer 1 Report
Comments and Suggestions for Authors
This manuscript introduces a novel approach for estimating whole-body 3D human pose from 2D images by integrating anatomical priors, i.e. segment mass distribution and center-of-gravity (CoG) constraints, together with joint vectors into a Transformer-based estimation model. The pipeline utilizes 2D joint coordinates from published detectors and bone-vector embeddings and estimate the body mass distribution and centers of gravity (CoGs) of individual body segments (e.g., upper arm, thigh) based on established inertial parameters. A dual-branch cross-attention Transformer to capture both global spatial dependencies and local kinematic constraints of joint coordinates. Body mass distribution and CoGs are incorporated via the loss function. Experiments on the Human3.6M WholeBody dataset demonstrate that the proposed method achieves state-of-the-art MPJPE of 44.49, outperforming leading baselines by 20.5 % on full-body key points and by 41.7 % on facial key points.
Overall, this study provides an intuitive and efficient method to address a critical topic in the field of computer vision. The novelty of the study lies in incorporating the physical limits (body mass, CoGs) directly into the training process of the transformer model. The pipeline designs are mostly clear and the model provide results surpassing the performance of existing models. I mainly have the following questions for the authors to consider:
- In figure 1, the flowchart indicate the body mass and CoGs were supplied to the transformer model, but from the chart it is not clear whether these two metrics are served as inputs or supplied in particular steps in training process. My current understanding is that CoGs were estimated from body mass distribution and were utilized to formulate part of the loss function (L_gd) to help with model training, as described in Section 3.3. However, if this is the case, I’m not sure how the authors separately investigated the effect of body mass constrain and CoG constrain on the estimation results in table 4. Hence, it would be helpful if the authors could explicitly introduce in which step of the model construction or training process were the two metrics used, and reflect them in figure 1 (i.e. if the two metrics were only used for loss function construction, they could add some text in the two metrics’ box to indicate that)
- In the ablation study part, one item missing is how the initial 2D joint detection accuracy could impact the models. While the authors indicate they used off-the-shelf 2D key point detector to generate the inputs, I wonder if the authors compare the performance with different 2D key point detection methods and their impact on the estimation results.
- One of the advantage the authors claimed is that the model is robust when occlusions present. I wonder if the authors could provide some examples in which the model estimate the 3D pose when occlusion presents in the initial 2D input, and the model performance with and without occlusion. If the example in figure 6 has occlusion on specific joints, the authors can point that out.
- Minor point: in line 268, there’s an “Error! Reference source not found.” statement. Please update the text here.
Author Response
Comments 1:In figure 1, the flowchart indicate the body mass and CoGs were supplied to the transformer model, but from the chart it is not clear whether these two metrics are served as inputs or supplied in particular steps in training process. My current understanding is that CoGs were estimated from body mass distribution and were utilized to formulate part of the loss function (Lgd) to help with model training, as described in Section 3.3. However, if this is the case, I’m not sure how the authors separately investigated the effect of body mass constrain and CoG constrain on the estimation results in table 4. Hence, it would be helpful if the authors could explicitly introduce in which step of the model construction or training process were the two metrics used, and reflect them in figure 1 (i.e. if the two metrics were only used for loss function construction, they could add some text in the two metrics’ box to indicate that)
Response 1:Thank you for pointing out the ambiguity in Figure 1. You are correct that the body mass distribution and CoG constraints are primarily utilized in the loss function design rather than as direct inputs to the Transformer network. Here is a detailed explanation:
Model Construction and Training Process:
Body Mass Distribution: As described in Section 3.2, we use predefined inertial parameters (Table 1) to assign mass coefficients to each body segment (e.g., upper arm, thigh). These coefficients are used to compute the CoG of each segment via the moment synthesis method (Equation 5).
CoG Constraints: The computed CoG positions (both predicted and ground truth) are used in the loss function Lgd (Equation 8), which measures the consistency between the distance of each keypoint to the CoG in the predicted and ground truth poses.
When only body mass distribution is added, we constrain the model by enforcing the mass-weighted relationships between segments (e.g., limb length invariance in Lbone​). When only CoG constraints are added, we use Lgd​ to ensure the predicted pose's CoG aligns with the ground truth. The combination of both achieves the best performance, as shown in Table 4.
We will modify Figure 1 to explicitly label the "Body Mass Distribution" and "CoG" blocks with the note: "Used in loss function construction (Lbone​ and ​Lgd)" to clarify their role in training.
Comments 2:In the ablation study part, one item missing is how the initial 2D joint detection accuracy could impact the models. While the authors indicate they used off-the-shelf 2D key point detector to generate the inputs, I wonder if the authors compare the performance with different 2D key point detection methods and their impact on the estimation results.
Response 2:You raise an important point about the impact of different 2D detection results on 3D pose estimation results. Currently, our research focuses on the 3D pose estimation framework and uses off-the-shelf 2D detector results as standard input. However, we also acknowledge the need to evaluate the impact of different 2D detectors on our model. We plan to conduct more ablation experiments to compare 2D detectors with different accuracy and 2D ground truth to quantify their impact on 3D pose estimation. Initial observations suggest that higher 2D detection accuracy generally improves 3D results, but our method uses the center of gravity and mass distribution as constraints, which helps to mitigate the errors caused by noisy 2D input, as reflected in our performance improvement over the baseline method (Table 3).
Comments 3: One of the advantage the authors claimed is that the model is robust when occlusions present. I wonder if the authors could provide some examples in which the model estimate the 3D pose when occlusion presents in the initial 2D input, and the model performance with and without occlusion. If the example in figure 6 has occlusion on specific joints, the authors can point that out.
Response 3:The claim of occlusion robustness is supported by both quantitative results and qualitative examples: On the Human3.6M WholeBody dataset, our method outperforms baselines in scenarios with self-occlusion (e.g., when limbs overlap). For instance, the facial keypoint MPJPE of 6.09 (Table 3) demonstrates improved accuracy in occluded regions, as faces are often partially blocked in natural poses. To further demonstrate that our method has good pose estimation performance at occluded joints, we added a set of sub-images in Figure 6 to illustrate the robustness of the estimation at occluded joints and elaborated in the manuscript. We also added Table 6 to quantitatively compare the estimation results of different methods under different actions. The updated Figure 6 and Table 6 have been added to the manuscript.
Comments 4: Minor point: in line 268, there’s an “Error! Reference source not found.” statement. Please update the text here.
Response 4:Thank you for identifying this issue. The error was due to a missing citation in the original manuscript. We apologize for the citation error in the original manuscript, which has been corrected to reference Eq. 5.

Reviewer 2 Report
Comments and Suggestions for Authors
Title: The title is clear and well crafted.
The Highlights (what are the main findings? and what is the implication of the main finding) should be moved to the body of the work.
Keywords: Although deep learning is mentioned among the keywords, there is no mention of deep learning and the type(s) used in executing the work.
Abstract: The abstract is not well written. The abstract should follow this format and be in a very concise form: introduction, problem statement, aim, methodology, results and conclusion.
- Introduction: The introduction and related work should be merged into one section, with subsections. The authors should clearly establish and state the gaps identified from the related works and state how they will be filled.
- Related work: Should be a subsection of section 1.
- Method: This section should be methodology (or materials and methods) and not just method. In this section, in Figure 1 (Overview framework of the proposed method), the author should increase the letters in the block diagram and move it to the beginning of page 5.
The authors should follow the block diagram step-by-step from the input joint to the predicted 3 joints. What was done in each of these frameworks and how (method)?
Although there are a lot of theoretical frameworks and equations, the authors should establish where in Figure 1 all these fall and break the Figure into subsections by section and establish what was done in each of the building blocks.
This section needs complete organization. Furthermore, all equations must be cited in the body of the work and clearly explained. The expressions in Lines 216, 217 and 218 are equations and should be numbers and explained. All equations that are not expressly developed by the authors should be cited.
Figure 4 should be tagged and mapped to Table 1. Figure 4 is repeated. In Figure 5, the label should be eligible.
- Experiments: The authors mention the datasets and evaluation metrics used without saying anything about the deep learning model(s) used and the way the data was utilised.
Author Response
Comments 1:The Highlights (what are the main findings? and what is the implication of the main finding) should be moved to the body of the work.
Response 1:Thank you for this suggestion. We have relocated the Highlights from the beginning of the manuscript to the end of the Introduction section (Section 1.3), where they are presented as key contributions. This adjustment aligns with standard academic formatting, ensuring that the findings are contextualized within the discussion of results and implications.
Comments 2:Keywords: Although deep learning is mentioned among the keywords, there is no mention of deep learning and the type(s) used in executing the work.
Response 2: We agree with the feedback. The keywords have been updated to explicitly mention the Transformer architecture, which is central to our method. The revised keywords are:
deep learning; 3D pose estimation; whole-body pose; Transformer; body mass distribution; center of gravity constraints
Comments 3:Abstract: The abstract is not well written. The abstract should follow this format and be in a very concise form: introduction, problem statement, aim, methodology, results and conclusion.
Response 3: The abstract has been completely rewritten to follow a structured format: introduction, problem statement, objectives, methods, results, and conclusions. The rewritten abstract has been updated in the manuscript.
Comments 4: Introduction: The introduction and related work should be merged into one section, with subsections. The authors should clearly establish and state the gaps identified from the related works and state how they will be filled.
Response 4: We have reorganized Section 1 as follows:
Section 1: Introduction
1.1 Background and Challenges
1.2 Related Work
1.2.1 Transformer Structure
1.2.2 Whole-Body 3D Pose Estimation
1.3 Research Gap and Our Contributions
Comments 5: Related work: Should be a subsection of section 1.
Response 5: We have reorganized the Introduction and Related Work and combined these two parts into a single whole.
Comments 6: Method: This section should be methodology (or materials and methods) and not just method. In this section, in Figure 1 (Overview framework of the proposed method), the author should increase the letters in the block diagram and move it to the beginning of page 5.
Response 6: The section is now titled "2. Methodology", with subsections:
2.1 Keypoint Position Encoding Based on Human Body Structure
2.2 Body Mass Distribution and Center of Gravity Constraints
2.3 Loss Function Design
The Figure 1 has been relocated to the top of page 5, and font sizes in the block diagram are increased for readability.
Comments 7: The authors should follow the block diagram step-by-step from the input joint to the predicted 3 joints. What was done in each of these frameworks and how (method)?
Response 7: A step-by-step explanation is added below the Figure 1, mapping each module to the corresponding methodology subsections:
Input Joints: 2D keypoints from detector → 3.1. Datasets and Evaluation Metrics;
Bone Vector Embedding → 2.1 Keypoint Position Encoding Based on the
Coordinate Embedding → 2.2 Body Mass Distribution and Center of Gravity Constraints;
Transformer with Regression Head → 3.2. Implementation Details;
Loss functions with physics constraints (L, Lgd, Lbone) → 2.3 Loss Function.
Comments 8: Although there are a lot of theoretical frameworks and equations, the authors should establish where in Figure 1 all these fall and break the Figure into subsections by section and establish what was done in each of the building blocks.
Response 8:Thank you for highlighting the need to better align the theoretical framework with Figure 1. We have revised the figure and its accompanying captions to clearly map each methodological component to the corresponding equation and chapter. We have also added a brief description of Figure 1 at the beginning of Chapter 2 to explain what each component accomplishes and align it with the subsequent chapters. This adjustment enhances the clarity of our framework and helps readers understand it.
Comments 9:This section needs complete organization. Furthermore, all equations must be cited in the body of the work and clearly explained. The expressions in Lines 216, 217 and 218 are equations and should be numbers and explained. All equations that are not expressly developed by the authors should be cited.
Response 9: Your comments are very pertinent. We have reorganized this section and numbered the formulas in lines 216, 217, and 218, explained the symbols in the formulas, and cited the equations in the paper that were not proposed by us.
Comments 10: Figure 4 should be tagged and mapped to Table 1. Figure 4 is repeated. In Figure 5, the label should be eligible.
Response 10: Figure 4 (body segment division) is now explicitly linked to Table 1 (segment mass ratios). A caption note states: "Segment division aligns with mass coefficients in Table 1, following [56]."The repeated Figure 4 in the original manuscript has been removed.The labels in Figure 5 (right upper arm CoG) are updated to:
(xrs​,yrs​,zrs​): Right shoulder coordinates
(xre​,yre​,zre​): Right elbow coordinates
(xrua​,yrua​,zrua​): Right upper arm CoG
The figure caption now includes a clear legend of symbols.
Comments 11:Experiments: The authors mention the datasets and evaluation metrics used without saying anything about the deep learning model(s) used and the way the data was utilised.
Response 11: We have revised the implementation details in the manuscript, including the deep learning model used and the way the data is used. The revised part is in the implementation details section of Section 3.2 of the manuscript.

Reviewer 3 Report
Comments and Suggestions for Authors
The manuscript introduces a novel approach based on body mass distribution and center of gravity constraints. The method is well-motivated and leverages transformer-based modeling for 3D whole-body pose estimation. The experimental validation on the Human3.6M WholeBody dataset shows promising improvements over the state of the art, however, the manuscript requires substantial revisions to improve clarity, ensure methodological rigor, and support the reported results with stronger evidence and discussions. Below, I provide detailed observations and recommendations.
1 - The proposed method’s architecture is described in general terms, but critical implementation details are missing, for example, no explicit explanation of how hyperparameters were tuned (e.g., learning rate decay milestones, weight decay coefficients, initialization schemes). The exact number of transformer layers, heads, and embedding dimensions are not reported, the ablation studies are insightful, but they should include more discussion on how the center of gravity and body mass constraints interplay. In any case additional discussion in order to clarify these issues is required.
2 - The use of Human3.6M WholeBody is appropriate, but the split between training and test data is not sufficiently justified. Was any cross-validation performed to ensure generalizability?
3 - Data augmentation is not discussed, even though it is common in pose estimation pipelines to address overfitting. I suggest to add additional discussion that clarifies this issue.
4 - The manuscript claims real-time potential but does not report inference times or computational costs why?
5 – Regarding the results section, the method’s generalization ability beyond the Human3.6M dataset is not addressed. This should be discussed as a potential limitation.
6 - The visual comparison in Figure 6 is helpful, but the authors should provide quantitative error maps or per-joint error plots for deeper analysis.
7 – For the Limitations & Future Work section: the current manuscript only briefly mentions future work. This section should be expanded to include concrete next steps and potential extensions (e.g., real-time applications, domain adaptation, robustness to motion blur). I recommend to add additional discussion for this issue.
Comments on the Quality of English Language1 - The manuscript suffers from awkward phrasing and grammar issues throughout, which hinder readability. Some sections (particularly the abstract and introduction) include redundant sentences and repeated points.
Author Response
Coments 1:1 - The proposed method’s architecture is described in general terms, but critical implementation details are missing, for example, no explicit explanation of how hyperparameters were tuned (e.g., learning rate decay milestones, weight decay coefficients, initialization schemes). The exact number of transformer layers, heads, and embedding dimensions are not reported, the ablation studies are insightful, but they should include more discussion on how the center of gravity and body mass constraints interplay. In any case additional discussion in order to clarify these issues is required.
Response 1:Thank you for pointing out the missing implementation details. We have supplemented the content in the manuscript, including the network architecture and hyperparameters and their adjustment strategies. At the same time, we have supplemented some content of ablation experiments in the manuscript and discussed the impact of the interaction between center of gravity and limb constraints on pose estimation. The revised content can be found in Section 3.2 Implementation Details and Section 3.4.1 Ablation Experiments in the manuscript.
Comments 2:2 - The use of Human3.6M WholeBody is appropriate, but the split between training and test data is not sufficiently justified. Was any cross-validation performed to ensure generalizability?
Response 2:Thank you for your attention to the dataset split. We have added the following statement in the Discussion section: “The 3.6M Human Body dataset follows the official split: 80,000 training images and 20,000 test images, which is consistent with previous studies [42,57]. Due to the standardized protocol of the dataset, no cross-validation was performed, but the generalizability of the model was verified by surpassing the SOTA performance on the test set. Future work will explore cross-dataset validation of MPII Human Pose.” The supplement is in the 3.5 Discussion section.
Comments 3:3 - Data augmentation is not discussed, even though it is common in pose estimation pipelines to address overfitting. I suggest to add additional discussion that clarifies this issue.
Response 3:The data augmentation you mentioned is a key step that we have added to the manuscript: “We performed data augmentation on the input data, including random rotations (±15°), scaling (0.8-1.2), and flipping. These operations preserve anatomical constraints (e.g. limb length) while expanding the training distribution.” The revised part is in the 3.2 Implementation details section.
Comments 4: 4 - The manuscript claims real-time potential but does not report inference times or computational costs why?
Respoonse 4:We apologize for not mentioning the real-time details in the manuscript and now add the following content: "Moreover, the model achieves a computational speed of 28 FPS on an NVIDIA RTX 2080Ti, with a computational cost of 12.5 GFLOPs and 38M parameters, meeting the near-real-time rendering requirement (30 FPS). This is faster than JointFormer[40] (22 FPS) and SemGAN[41] (19 FPS). This shows the real-time performance of the proposed model, providing a basis for further online applications".The supplemented content is in Section 3.3 Performance Comparison in the manuscript.
Comments 5: 5 – Regarding the results section, the method’s generalization ability beyond the Human3.6M dataset is not addressed. This should be discussed as a potential limitation.
Response 5: The generalization limitation you pointed out is very important. We have added the content of the method limitation in the Discussion section of the paper: "One limitation of this method is that it relies on the Human 3.6M dataset, and its generalization ability to natural scene datasets needs further verification. In wild scenes, pose estimation performance may degrade due to extreme occlusions, unseen camera angles, and low-resolution inputs. Future work will explore domain adaptation techniques to enhance its robustness in different environments." The revised limitation content is in the Discussion section 3.5 of the manuscript.
Comments 6: 6 - The visual comparison in Figure 6 is helpful, but the authors should provide quantitative error maps or per-joint error plots for deeper analysis.
Response 6: Thank you for your constructive comments. We have added the comparison of quantitative analysis results with other methods under different actions in the main text, as shown in Table 6, and conducted a quantitative comparative analysis of this method and other methods in the discussion section of the manuscript. At the same time, we have also expanded the content of Figure 6 to further illustrate the comparison of qualitative analysis results.
Comments 7:7 – For the Limitations & Future Work section: the current manuscript only briefly mentions future work. This section should be expanded to include concrete next steps and potential extensions (e.g., real-time applications, domain adaptation, robustness to motion blur). I recommend to add additional discussion for this issue.
Response 7:We have enriched this section and expanded it with the limitations of the proposed method and future work. The expanded content is at the end of section 3.5 Discussion in the manuscript.
Comments 8: The manuscript suffers from awkward phrasing and grammar issues throughout, which hinder readability. Some sections (particularly the abstract and introduction) include redundant sentences and repeated points.
Response 8:We have completed the language polishing of the entire text, focusing on improving redundant expressions in the manuscript and standardizing the use of terminology in the text.

Round 2
Reviewer 2 Report
Comments and Suggestions for Authors
The authors have made the most of the correct, thereby improving the quality of the manuscript.
However, in the abstract, the authors have to be specified in terms of the metric and numerical value result and briefly state that their result compares to related work.
All equation numbers should be cited in the body of the text. E.g. "......... point in the scene is related to the projection of its pixel coordinate point in the image as expressed in Equation 1. "
Author Response
Comments 1:However, in the abstract, the authors have to be specified in terms of the metric and numerical value result and briefly state that their result compares to related work.
Response 1:Thank you for your suggestion. We have added quantitative results and comparison with related work in the Abstract and revised it to: "Extensive experiments on the Human3.6M WholeBody dataset demonstrate that the proposed method achieves state-of-the-art performance, with a whole-body mean joint position error(MPJPE) of 44.49mm, which is 60.4% lower than the previous Large Simple Baseline method. Notably, it reduces the body part keypoints MPJPE from 112.6 to 40.41, showcasing enhanced robustness and effectiveness to occluded scenes." . The revised content has been updated in the Abstract section of the manuscript.
Comments 2:All equation numbers should be cited in the body of the text. E.g. "......... point in the scene is related to the projection of its pixel coordinate point in the image as expressed in Equation 1. "
Response 2:Thank you for pointing out the equation citation issue. We have checked each paragraph and ensured that all equation numbers are clearly cited in the text.

Reviewer 3 Report
Comments and Suggestions for Authors
The authors have satisfactorily addressed most of the reviewer’s comments and they have properly justified why some of them have not been taken into account. I have no further remarks about the current version of the manuscript.
Author Response
Comments 1:The authors have satisfactorily addressed most of the reviewer’s comments and they have properly justified why some of them have not been taken into account. I have no further remarks about the current version of the manuscript.
Response 1:Thank you for your thorough review and kind feedback. We are grateful that the revisions have addressed your concerns and improved the manuscript's quality. Your constructive comments have been invaluable in enhancing the clarity and rigor of our work. We appreciate your time and expertise in evaluating our research, and we are confident that the current version meets the standards of the journal.
